# Financial Analysis and Survival Research of the Visegrad Countries' Health Industries

**Judit Vitéz-Durgula** [1], **Anna Dunay** [2], **Gergő Thalmeiner** [3,*], **Balázs Vajai** [1] **and László Pataki** [2]

1   István Széchenyi Economics and Management Doctoral School, Alexandre Lamfalussy Faculty of Economics, University of Sopron, 9400 Sopron, Hungary; durgula.judit@phd.uni-sopron.hu (J.V.-D.); vajai.balazs@phd.uni-sopron.hu (B.V.)
2   Hungarian National Bank—Research Center, John von Neumann University, 6000 Kecskemét, Hungary; dunay.anna@nje.hu (A.D.); pataki.laszlo@nje.hu (L.P.)
3   Department of Investment, Finance and Accounting, Hungarian University of Agriculture and Life Sciences, Páter Károly Str. 1, 2100 Gödöllő, Hungary
*   Correspondence: thalmeiner.gergo@uni-mate.hu

**Abstract:** Our study examined the financial situation of small and medium-sized enterprises (SMEs) in the health industry of the Visegrad Four (V4: Czech Republic, Poland, Slovakia, Hungary) in the period 2008–2021. The basis of the analysis was the reports available in the Crefoport database. During the analysis, we took into account four key financial indicators (liquidity, profitability (ROA), efficiency, capital structure) and used the Kaplan–Meier survival estimator to examine the viability of enterprises. In our study, we draw attention to the national economic importance of healthcare SMEs in the CEE region, and within that, in the V4 countries during the period of Industry 4.0 and the COVID-19 pandemic. Our research goal is to examine the life expectancy of healthcare enterprises operating in Hungary at the time of their establishment, in relation to the SMEs of the other three countries. The health industry SMEs of the V4 countries showed positive growth (+18%) in the period under review compared to the average of the 3 years before the COVID-19 pandemic, in the average of the first 3 years of the COVID-19 pandemic. The study paid particular attention to the life expectancy of businesses at the time of establishment. According to our results, the life expectancy of healthcare enterprises operating in Hungary at the time of establishment is high, but it does not differ significantly from that of the other examined countries. Our study highlights that the health industry SMEs of the examined countries do not need financial support; they are standing on a stable footing, which provides them with an excellent opportunity to either expand abroad or introduce innovations.

**Keywords:** SME; V4 countries; health industry; financial analysis; survival analysis

## 1. Introduction

The proper functioning of the health industry is essential for the users, providers, managers, financiers of the health service, and the actors of the industries that serve the health services [1]. By the health industry, many people mean healthcare and the healthcare system, and by the corporate side, the pharmaceutical industry (and the companies that represent it) [2–4] and biotechnology startups [5–7] are the popular research areas. The policy importance of healthcare and the healthcare system is unquestionable, but healthcare SMEs receive less emphasis, even though their importance is indisputable and they form an integral part of the healthcare industry [8]. The SME sector of the health industry has been investigated in a lesser degree, which is why its mapping may contain novelties.

In the last 30 years, since the Visegrad Group (V4) exists, the member countries (Poland, Hungary, Slovakia, and the Czech Republic) have undergone radical changes. Overall, it can be said that the population has become relatively healthier, wealthier, and more educated, and the economies are operating more efficiently. However, during the appraisal of a country's prospects, rankings are not the only things to be considered; they can be

misleading. It is important that the reforms do not only work on paper, but also generate added value and/or contribute to improving the living conditions of the population [9]. It is essential to get to know the problems with which the examined economies and the examined industry, the health industry, are struggling; but, we also present the ideas and characteristics that can provide an additional competitive advantage for the prosperity of SMEs in the health industry and provide a more livable place for society.

In our study, we focus on highlighting the role of healthcare SMEs in the national economy in the Central and Eastern European (CEE) region, especially in the V4 countries, where the Industry 4.0 paradigm shift and the COVID-19 pandemic had a serious impact on the operations and financial situation of businesses. In this context, the aim of our research is to examine whether the life expectancy of healthcare enterprises operating in Hungary at the time of establishment is longer than that of the enterprises of the other three countries. In the course of our research, we point out the importance of healthcare SMEs in the CEE region, showing that these enterprises are not only the driving forces of economic development, but also create important value for society. The economic analysis of the healthcare SMEs of the V4 countries and the investigation of their sustainable operation is an important and timely research area, the understanding of which is essential to promote the long-term stability and growth of the region's healthcare sector and economy. Healthcare SMEs contribute to the national economy, promote innovation, and play an important role in improving society's healthcare. The application of mathematical models and economic indicators in the evaluation of company lifetime, profitability, competitiveness, and sustainability is not only scientifically exciting, but also contributes to the theoretical and practical understanding of the field. The detailed insight provided by the analyses enables the discovery of systemic risks and opportunities, the support of economic policy decision making, and the development of industry innovation strategies. Furthermore, the analysis of sustainable operations is extremely important in evaluating the long-term economic viability and social added value of SMEs. The sustainable business models of healthcare SMEs play a central role in providing healthcare, increasing the well-being of society, and maintaining the economic stability of the region. So, the sustainable and viable management model followed by healthcare SMEs serves the long-term interest of society as a whole and needs a more in-depth analysis in future research.

This paper is organized as follows. Section 2 presents the data collection and associated methods. Section 3 presents the results, during which we present the analysis of financial indicators and the results of the Kaplan–Meier estimator. In Section 4, the discussion, previous research found in the literature that is related to our research is summarized. Section 5 presents the conclusions based on the results. Section 6 highlights the study's advantages, limitations, and future research opportunities. In Section 7, the implication of the study is described.

## 2. Materials and Methods

### 2.1. A Brief Introduction of the Visegrad Countries

The Visegrad Group was created as an informal alliance of four countries. They were connected by proximity and similar geopolitical conditions, as well as a common history, tradition, culture, and group of values [10]. The V4 countries are currently the sixth largest economic power and the third largest consumer market in Europe. The enlargement of the European Union to include the V4 countries embodies all their aspirations. The V4 performance ended the split of Europe into two rival blocs and gave the new member states a chance to catch up with their Western counterparts, by modernizing their physical infrastructure and economy, for which the EU provided the resources. Without this solidarity and support, the process would have been much longer, if it was realized at all. From this point of view, the EU membership of the countries of the Visegrad Group is undoubtedly a success story [11]. The low innovation potential of the Visegrad Group compared to the EU average affects their weak economic position compared to the more developed countries. This distance has only deepened over the last years. Examples include

indicators such as enterprises' expenditure on research and development, participation in lifelong learning, or employment in the high-tech sector. Two of the few potentials that can determine the economic competitiveness of the V4 Group in the future are the human capital and the high level of education of the society. As a result of the analysis, it was also revealed that the most innovative potential of the Visegrad Group countries lies in the Czech and Hungarian regions [12]. The Visegrad countries are trying to become more competitive by attracting external resources (capital and technology) against their more developed rivals. This is favored and supported by low tax rates, low labor costs, and decreasing legal and administrative burdens [13]. It would be desirable to create and expand the R + D + I cooperation opportunities of the V4 countries (on scientific, research, and business levels, as well), since companies and clusters based on high-tech technology could be created, which would also increase the competitiveness potential of the Central and Eastern European region [14]. Visegrad countries, a place between developed Europe and Eastern Europe, are a mixture of two worlds that have successfully transformed from communism to capitalism. In order to shape a successful future, the help of all stakeholders will be needed due to institutional gaps, political resistance, and lack of knowledge and financial resources. Taking into account the lessons learned from the transformation of the past decades, the first steps have been taken on the road to future prosperity, but there is still much to be accomplished [9]. Since the Russian–Ukrainian conflict, the differences among the member countries have intensified, and constructive cooperation works in few areas. In its current state, cracks are visible in the V4 cooperation.

### 2.2. Overview of the Health Industry in the Visegrad Countries

Research topics related to the health industry can be extremely diverse and ramifying. The health industry is a complex field that includes many aspects; here are the most frequently researched fields from the economic perspective: health technologies [15,16], personalized health [17], e-health [18,19], big data [20,21], artificial intelligence [22,23], health efficiency and financing of systems [24,25], and the impact of the COVID-19 pandemic and building a resilient healthcare system [26].

We have collected studies about the health situation of the Visegrad countries for further examination. Bohle-Greskovits (2012) grouped the post-socialist capitalist systems into neoliberal (Baltic countries) and embedded neoliberal (Visegrad states) types [27]. The Visegrad Group combined radical marketization with minimal social protection, while the Baltic States compensated the losers of marketization with more generous welfare conditions. Zatonski (2007) documents that in the post-socialist Central and Eastern European countries, adult mortality rates began to differ from Western countries in the 1960s, and this is where the health gap between Eastern and Western Europe began [28]. Hejduková–Kureková (2017) examined the health performance of the V4 countries and found several common points: the Central and Eastern European countries are transitional economies, and the health status of the population is worse than in the more developed countries of the OECD or the EU; healthcare systems are underfunded; reforms serve only economic interests; and there is a lack of a complex health strategy from the point of view of long-term sustainability [29]. The large proportion of the population that experienced stress, financial difficulties, and job loss due to the change of regime contributed to the current health disadvantage of the Central and Eastern European region [30].

There are several methods for examining the healthcare system, and researchers are constantly creating new ones. At the same time, several researchers remark that when measuring the performance of care systems, without measuring the inputs and outputs, it is difficult to determine the impact of a healthcare reform and to verify whether it has achieved its goal [29]. Hejduková–Kureková (2017), based on their indicator containing ten variables (their data refer to the OECD Health Policy study between 2005 and 2014), determined that the best care system is in the Czech Republic, followed by Slovakia and Poland, and then Hungary in the last position [29]. The relative efficiency of the healthcare system can be analyzed using the DEA (Data Envelopment Analysis) method, too. As a

result, in the period of 2004–2010, the Polish healthcare system was classified as one of the most efficient, while the other three countries were found to be operating inefficiently, with an unbalanced rate of development. According to this, the Czech Republic maintains a stable position, while in the cases of Slovakia and Hungary, there are fluctuations—they did not even reach the limit of efficiency [31].

Ferreira et al. (2018) examined the European Union healthcare systems (based on the three health system functions proposed by the WHO: care, resource supplies, financing) using multivariate analysis methods based on data from 2012 (or the latest available year). It has been proven that these countries have the lowest life expectancy at birth and low GDP per capita [32]. This situation report is also supported by more recent research between 2013 and 2018 in the context of healthcare systems. These countries have the worst health conditions and availability of hospital beds, as well as the average amount of medical expenses and satisfaction with the care system [33]. Paulikné (2019) presents and evaluates the healthcare systems of the Visegrad countries. According to her comparison, Hungary and Poland are the back makers, while the Czech Republic and Slovakia show an improving tendency [1].

### 2.3. Characteristics of the SME Sector of the Visegrad Countries

The characteristics of the SME sector of the Visegrad countries represent a popular research area [34–38]. Berger-Udell (1998) defined three factors that influence the financing possibilities of an economic organization: the size of the enterprise, its age, and the available information [39]. The industry sector in which the enterprise or the company operates will affect its financial performance. Differences in performance may arise from financial and/or operational decisions. There is a relationship between innovation performance (e.g., product, service, and process innovation) and corporate financial performance (e.g., market share, sales volume, profitability). The literature shows a correlation between the age of the firm and the financial performance: older firms perform better than younger ones, while young firms are more agile and innovative, resulting in higher sales volume. In addition, financial performance decreases as the business grows older, but increases in the case of start-ups [40]. In terms of their number and importance, SMEs play a decisive role in creating a sustainable future through responsible business practices. Responsible business practice is about continuous improvement through the integration of the basic idea of sustainability into the organization, mission, and strategy. Sustainability can be considered a critical factor, with a view to the future viability of SMEs [41]. The study of Dvoulety et al. (2021) aims to contribute to the entrepreneurship research from the perspective of Central and Eastern Europe. They found that entrepreneurial returns in V4 countries are 22% higher on average than the employee return [42]. Kézai-Kurucz (2023) examined resilience, which is essential for startups in the V4 countries, as a key to survival during the COVID-19 pandemic [43].

### 2.4. The Relationship between Innovation and Sustainability in the Health Industry and the SME Sector

The sustainability of the health industry requires innovation. In the field of health science and health industry, hospitals and clinics play distinguished roles in innovation, which appear as key players in the application, reproduction, and creation of medical and new knowledge. After all, these institutions provide the essential part of healthcare and integrate and use new technologies, in other words, they represent a demand for innovation [44].

SMEs are willing to take the risk to be at the forefront of innovations in the health industry, but this requires national and European Union policy efforts and support in order to be able to offer solutions to the ongoing health challenges. Healthcare SMEs mainly offer the following healthcare products and services: medical equipment, instruments, and services, as well as biotechnology, operation of diagnostic laboratories and production of materials, production of primary prevention sensors, and drug development and delivery.

In the case of innovative medical devices, there are enormous opportunities in personalized medicine, including diagnostics, therapy, and monitoring. SMEs also have a lot of potential in terms of general healthcare [45]. In Europe, many healthcare SMEs have been operating as family businesses for up to 75 years. These SMEs need a generation change; they need to find the ways to commercialize new and emerging technologies, or ways to apply existing technologies in the new environment. SMEs mainly operate within countries, as relatively few of them carry out cross-border business activities, although this also applies to healthcare SMEs, which should participate in cross-border cooperation—the policy summary document writes it as a recommendation [45]. Innovation plays a major role in SMEs' competitiveness, sustainability, and employee performance [46]. Innovation supports the sustainable management of enterprises, which has a positive return not only in the environment, but also in society and entrepreneurial culture. The importance of the topic is shown by the increase of scientific research [47].

### 2.5. The Performance of the Health Industry of the V4 Countries, with Particular Regard to the SME Sector

In the next section, we present the latest statistics on the health industry of the V4 countries, focusing on the performance of the corporate sector. Europe's healthcare and biotech industry has been a pioneer for a long time in the field of development of medicines and treatments that save lives and improve people's health and well-being around the world. This is shown by the fact that 23% of all European venture capital investments, a total of EUR 13.5 billion, went to 16% of all companies that received venture capital, i.e., 2634 companies operating in the biotech and healthcare industries between 2017 and 2021. From this amount, 110 companies—in the CEE region—received a capital investment of EUR 0.09 billion in the period under review. There are few sectors where venture capital investments can have such a direct and beneficial impact on the lives of millions [48]. After the ICT sector, the most sought-after industry across Europe is healthcare and biotechnology, in which venture capital investors preferred to invest in 2022 [49]. The most promising areas of business angel investments were also healthcare and sustainable approaches in 2022 [50]. The European innovative healthcare SME sector is worth more than EUR 250 billion; in addition, the EU spends 10% of its GDP on healthcare and employs 17 million people in the industry [45].

According to the EISMEA (European Innovation Council and SMEs Executive Agency), in 2021, the health ecosystem represents a total of 10% of the added value of the SME sector measured in 14 areas (ecosystems) of the EU27 countries. The share of micro SMEs in the number of enterprises in the health industry is 93.5%. Examining total employment, micro SMEs represented 22%, small SMEs 15%, medium-sized SMEs 16%, and large enterprises 47% in 2021; together, they added a value of EUR 255.428 million. This is 5.2% of the total added value of the 14 ecosystems, including all corporate groups. The distribution of the added value of the healthcare ecosystem (EUR 255.428 million) by company size is the following: micro SMEs 9%, small SMEs 9%, medium-sized SMEs 11%, large enterprises 71%. The percentage change of the added value from 2020 to 2021 was the largest in the health industry, 14.2% in total (SMEs 8.1%, large enterprises 16.6%), presumably due to the COVID-19 pandemic. The share of the cumulative change of total value added between 2019 and 2021 attributed to SMEs by the health ecosystem was 3%. The healthcare industry had the least change of added value between 2019 and 2021 (3%). The growth rate of large companies (17%) was more than double the growth rate of SMEs (8%) for 2020–2021 [51]. It is important to note: missing NACE (Nomenclature of Economic Activities) sector Q (human health and social work activities). By way of comparison, the ratio of profitable companies examined in the EIBIS (European Investment Bank Investment Survey) is relatively stable; it fell sharply during the COVID-19 crisis (by 8 percentage points for SMEs and 6 percentage points for larger companies). Since then, all categories have recovered, but only partially, filling three-quarters of the gap caused by the crisis. In 2022, 80% of European companies were profitable, and the overall profit ratio far exceeds the pre-crisis level in

the CEE countries [50]. The number of enterprises in the health ecosystem and their size class with total share in 2021 is micro SMEs 460,230 (92.38%), small SMEs 30,287 (6.08%), medium-sized SMEs 5958 (1.2%), and large enterprises 1731 (0.35%) [51]. The number of small and medium-sized enterprises (SMEs) in the non-financial business economy of the European Union (EU27) in 2022, by country, is Hungary: 688,900; the Czech Republic: 1,048,629; Slovakia: 506,888; and Poland: 2,078,056 [52]. It can also be said about health industry SMEs that they are typically smaller in size, but have great potential in terms of innovation, flexibility, and quick response to the sector needs.

Figure 1 summarizes the aggregated performance of the knowledge-intensive, high added value-producing, primarily intellectual capital-based industries related to the health industry in the years before (2017–2019) and during (2020–2022) COVID-19, in the Visegrad countries.

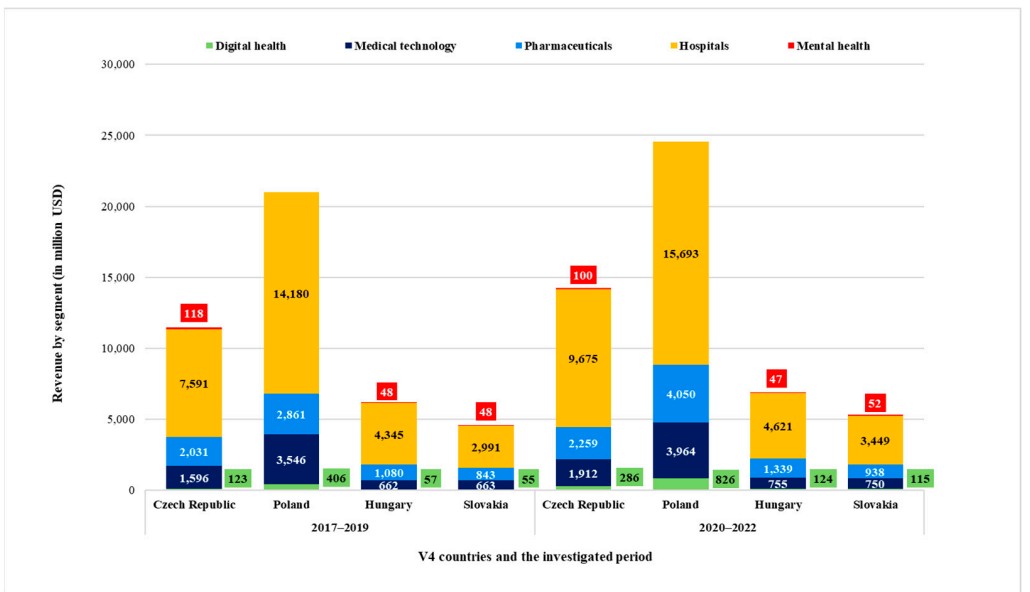

**Figure 1.** Sales revenue of the health industry of the V4 countries broken down by industry, before and during COVID-19. Source: Own editing based on [52].

It is important to note that the figure presented here is based on Statista's industry reports and methodology; it does not take into account, for example, the biotechnology industry, and there are no data on the cannabis industry, so it cannot be called comprehensive. At the same time, with its limitations, it can be considered a good approximation during the analysis of the health industry in the V4 countries. On the average of the period between 2017 and 2019, the overall sales revenue of the health industry by industry as interpreted by Statista was toweringly highest in Poland. By way of comparison, 47% of the total Polish healthcare industry sales revenue came from Czech healthcare companies, barely a quarter of Hungarian sales revenue, and 17% of Slovak sales revenue.

On the average of the COVID-stricken years (2020–2022), the overall sales revenue of the health industry was also the highest in Poland, representing a 17% increase compared to the previous period; the Czech Republic 24%; Hungary 11%; while Slovakia achieved a 15% increase. In terms of industries, the mental health industry was dwarfed (in fact, it shows a total decline of 7.5% during the examined periods) by the hospitals industry, which cut a huge slice. The digital health industry achieved the fastest growth—in the examined periods—by 211% in the aggregate of the V4s [53–60].

## 2.6. Methodology

There is an insignificant number of studies about the financial performance of the SME sector of health industry enterprises. We did not find studies that compared the enterprises of the industry on a global scale. Supplying the deficiency of the literature

connected to the subject is the aim of our research. To determine the multitude, we used the company information database of Crefoport Scholar (https://www.crefoport.hu, accessed on 14 February 2023), from which the data were collected by country along the following conditions:

1.  Conducting main activities by the scope of activities of the health industry [61–65]. Although the examined countries do not use the same main activity numbers, they can be roughly matched according to the global industry classification standard.
2.  Can be classified as SMEs. In international practice, the statistical delimitation of SMEs is conducted exclusively according to staff-category due to the lack of a consistent methodology [66]. The V4 countries, therefore, do not yet produce data based on the Hungarian terminology. In this research, for the sake of comparability, we applied the provisions of the Hungarian Act 2004. XXXIV. on supporting small and medium-sized enterprises and their development, which was written along the lines of EU recommendation No. 2003/361/EC [67].
3.  In the examined last 2 years (2020–2021), the company meets the SME criteria (employee number and balance sheet total or sales revenue)—in the amount of equivalent euros, using the average exchange rate of the European Central Bank (31 December 2021).
4.  To represent active, functioning social enterprises that have submitted an annual report for at least the years of 2020–2021.

A total of 20,817 enterprises met the above conditions in the Crefoport database on 14 February 2023, when 7035 Hungarian, 4657 Polish, 7608 Slovak, and 1517 Czech healthcare SMEs were filtered. The size of the starting database is, therefore, 20,817 elements. During the database cleaning, we used expectations that the value of equity is greater than 0, as well as that the value of all assets is greater than 0. During the management of extreme values per country, the top and bottom 5% were eliminated, and we examined this cleaned sample chronologically during 2008–2021 (or during the available period).

Aggregate data covering the entire sector were analyzed with the Microsoft Excel 2307 build version 16.0.16626.20086 program and the R 4.3.1x86 program. R is a language and environment for statistical computing and graphics.

We used four indicators and one estimator to examine the financial situation. Since it is a general problem to use an indicator with the same name with different content, or an indicator with the same content with a different name, in the following, we elaborate what we mean by the analyzed indicators.

*   Liquidity: By liquidity, we mean the company's solvency. With the help of liquidity indicators, it can be determined how well a company or business is able to meet its short-term obligations with the help of its current assets at a predetermined and previously agreed time, continuously and without delays [68].
*   Efficiency: Efficiency was calculated using the direct efficiency indicator: Sales revenue/Total assets. It shows how much net sales the company achieved in the given period by operating all the assets shown in its balance sheet [69].
*   ROA: Return on Net Assets = (Taxed profit/TOTAL assets) is known worldwide; it is a top profitability indicator that maps the return on assets, and for which "the bigger, the better" valuation principle is true [70,71].
*   Indebtedness: Among the capital structure indicators, we used one of the most frequently examined indicators, which indicates the ratio of Foreign capital/Own capital. (D/E) [72].
*   Kaplan–Meier estimator: The Kaplan–Meier estimator enables the description of survival probabilities over time. According to its structure, the model estimates the survival function of the entire population based on 2 pieces of information—the time elapsed until the target event and a binary variable describing the occurrence of the target event [73]. During the statistical analysis, we also perform the log-rank test. The log-rank test is a statistical test used for comparisons based on survival curves. The log-rank test is used to statistically evaluate whether there is a significant difference in survival between two or more groups. The test examines the difference between

expected and observed survival rates based on time-varying events. The log-rank test is based on the same assumptions as the Kaplan–Meier survival curve [74,75].

## 3. Results

### 3.1. Liquidity

Taking a closer look at the aggregated liquidity of the health industry enterprises representing the V4 countries included in the study (Figure 2A), it can be concluded that their liquidity is stable and balanced. Starting from 2008, the average liquidity takes a value of around 1.6 until 2012, then it starts to increase slightly, and from 2016, the average liquidity value exceeds 2.0, which already reflects a stable financial situation. The solvency of the middle 50% of the investigated enterprises is improving; it can be said that a liquidity surplus has arisen. This may cover an unreasonably conservative, cautious, inappropriate investment approach [71], which refers to deficiencies in management [76] regarding the use of inappropriate cash or other short-term assets. It would be worth investigating the reason for this, whether the stock of current assets increased to such an extent, or whether the drastic reduction of short-term liabilities resulted in this high ratio.

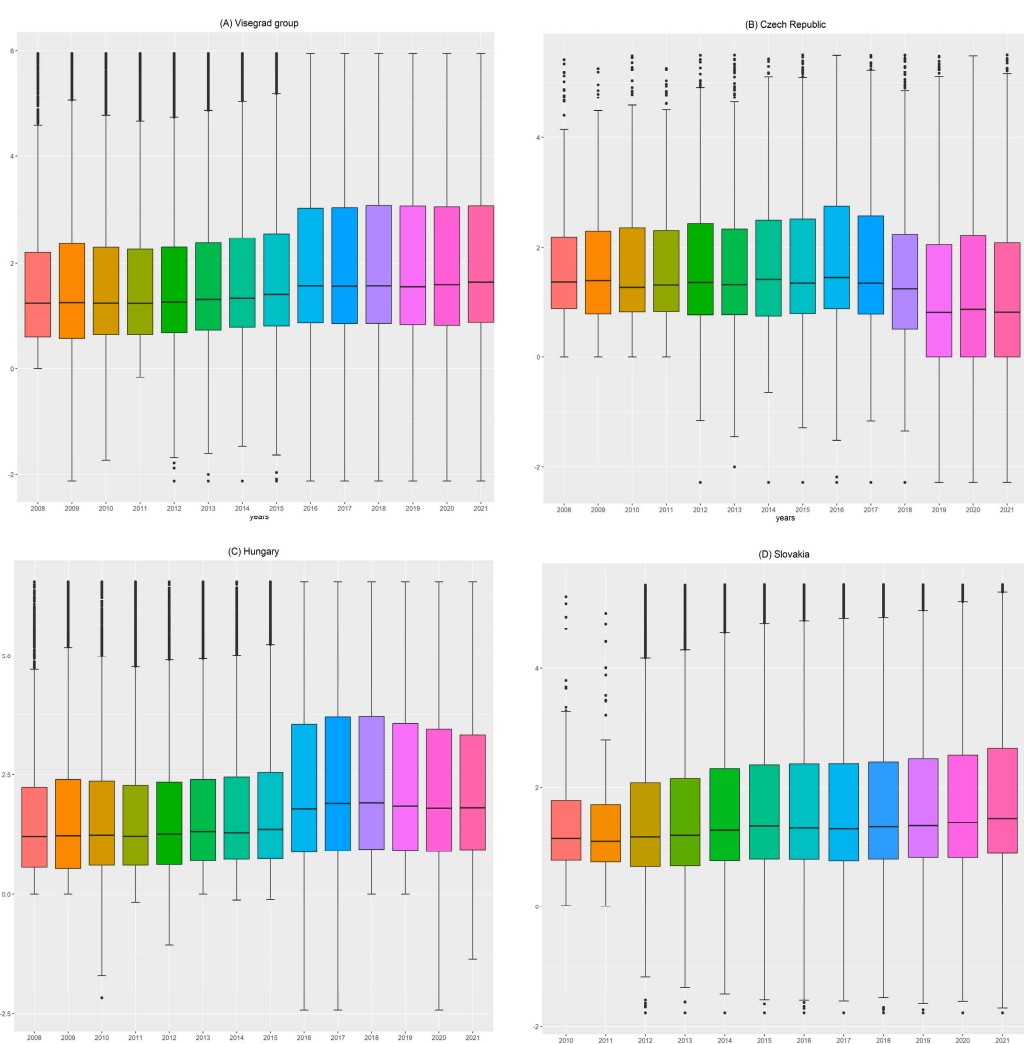

**Figure 2.** Liquidity boxplots of examined health industry SMEs of the V4 countries ((**A**) Visegrad Group, (**B**) Czech Republic, (**C**) Hungary, (**D**) Slovakia).

Examining individual countries, it can be said about the liquidity of Czech healthcare companies (Figure 2B) that the sample has a large standard deviation, the extremes are increasing, there are more outliers, and the overall average liquidity is decreasing over the

entire period under review; between 2008 and 2016, it increases to 1.6–1.8; between 2017 and 2021, it shows a strong downward trend, with a value close to 1.3, which can still be considered medium liquidity.

In the case of Poland, we can make more cautious observations; since the available time series are shorter, the examined period is limited to 2017–2021. The liquidity index is above 2 every year under review; moreover, it even reaches 10 in 2019–2020.

The diagram of the solvency of the examined Hungarian healthcare SMEs (Figure 2C) is very similar to the aggregated diagram (perhaps due to the high number of Hungarian elements). Each year under review, it can be said that there are more companies with better liquidity indicators than average or weaker ones. It is observable, between 2008 and 2015, that there are more extreme values, and the indicator moves in a relatively narrow band. Also in this period, half of the companies have acceptable liquidity, with a value close to 1.3, while from 2016, the proportion of those with a stable financial situation increases significantly. Looking at the examined period as a whole, it can be said of about 25% of the companies that a short-term insolvency of the companies may occur, because they take a value below 1. However, starting from 2016, half of the companies show liquidity above 1.8 (median).

The enterprises of Slovakia are analyzed based on reports made between 2010 and 2021. Figure 2D illustrates the financial stability of the examined health industry enterprises, which shows an average of over 1.3 in each year under review, and even 50% of them have a solvency ratio of 1.3 (starting from 2015; before that, it shows 1.1). This also supports a strong liquidity situation, as there are more companies with above-average liquidity in the sample.

Analyzing the solvency of the investigated companies of the V4 countries, we found that liquidity is high. It is also worth studying profitability, for which we chose the ROA indicator.

### 3.2. ROA

Looking at the overall figure of the V4s, the average of the ROA (Return on Net Assets) indicator ranges between 10 and 12% in the examined period. It is worth examining the indicators of each country, which show compared to the total average assets, how much profit after tax was achieved by the enterprises included in the study. A great advantage of the indicator is that it takes into account the indebtedness. The comparison is valid both chronologically and within the industry; the numbers obviously indicate which companies are performing better and what trends are emerging.

Czech data can be considered relevant from 2016. The average value of ROA shows an increasing trend until 7–10%. The examination of the profit ratio of the assets of the Polish enterprises can also be interpreted in a narrower interval, from 2017, when there are still a lot of outlier data. In 2020, it can already be said that 50% of the companies included in the study have an ROA value of 0, while in 2021, the middle 50% of companies produced an ROA between 0 and 22%. The individual numbers of the Slovakian and Hungarian samples show a very similar ROA value (with the exception of 2010–2011) in the examined period (Figures 2B and 3A). In Slovakia, the average value of the quotient is 10% from 2012, and half of the examined companies in the Slovakian health industry perform better than 9–11%, while the median value for the Hungarians is slightly higher, ranging between 10 and 13%.

Histograms can also be used to illustrate the frequency of different values, thus providing information about the distribution within the given variable sample. The X-axis shows the possible values of efficiency, and the Y-axis shows the frequency, i.e., the number of businesses that take the given value or fall into the given value range.

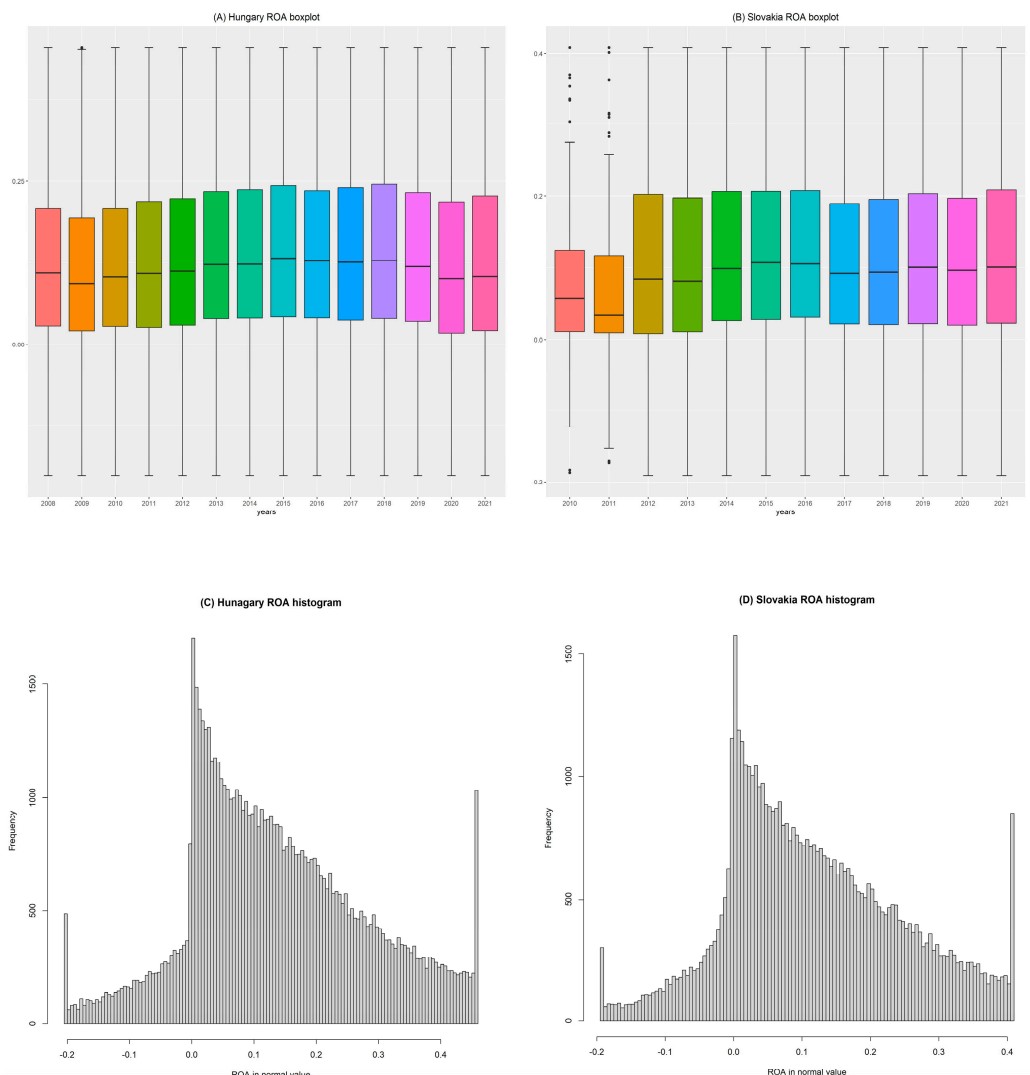

**Figure 3.** ROA indicator of the examined health industry SMEs of the V4 countries ((**A**) Hungary ROA boxplot, (**B**) Slovakia ROA boxplot, (**C**) Hungary ROA histogram, (**D**) Slovakia ROA histogram).

In the Hungarian and Slovakian samples, the location of the individuals is not symmetrical; the sample is positively skewed. The shape of the distribution of the sample shows positive pointedness, that is, the individuals are more clustered around the average (Figure 3C,D).

We can state that it is necessary to harmonize the liquidity risk of healthcare enterprises with the profitability situation, since maintaining sufficient liquidity supposes the profitable management.

### 3.3. Indebtedness

Among the capital structure indicators, we used one of the most frequently investigated indicators, which is the Debt/Equity ratio (D/E). The value of indebtedness shows a favorable trend in the aggregated D/E figure (Figure 4A), as the average indebtedness of the examined SMEs is decreasing: in the source structure, the value of the equity exceeds the value of external capital. Even the maximum values do not reach the value of 2, when the level of indebtedness is already very high.

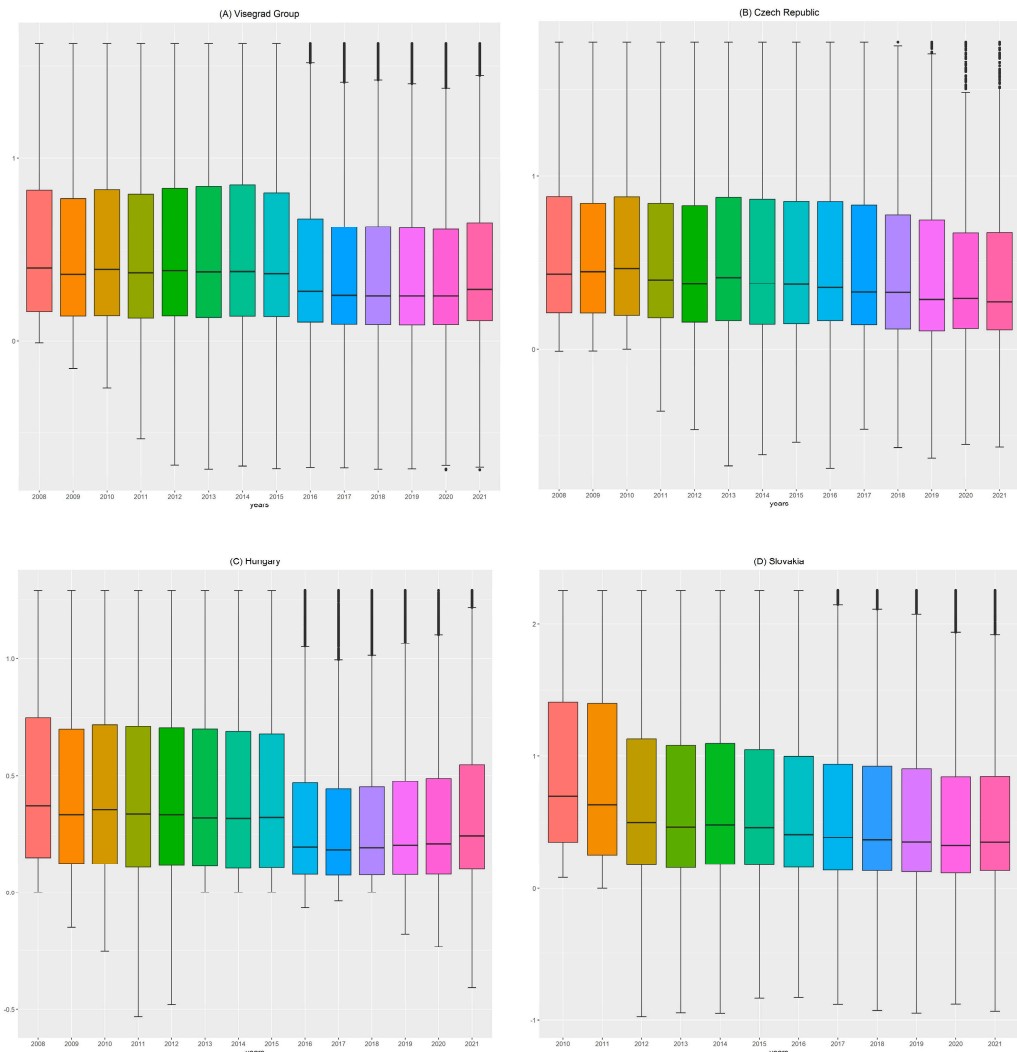

**Figure 4.** Indebtedness boxplots of examined health industry SMEs of the V4 countries ((**A**) Visegrad Group, (**B**) Czech Republic, (**C**) Hungary, (**D**) Slovakia).

The average of the indebtedness index of Czech SMEs (Figure 4B) during the time series is balanced at 0.5. In the years affected by COVID-19, the ratio of equity increased by a 6/4 ratio compared to the ratio of foreign capital. The favorable indebtedness ratio indicates that, in the case of an economic downturn, the SMEs included in the study will be able to cover their existing debts on their own. Of course, this may also indicate that they are not able/dare/want to take out more loans.

The indebtedness of the Hungarian sample can be considered the most favorable among the V4 countries (Figure 4C). It also shows a decreasing trend during the period; the initial average value decreased from 0.48 to 0.36 by the end of the period. A low D/E ratio means that the company relies on foreign sources for its financing in a relatively small part.

The Polish indebtedness does not show such a balance (the shortness of the time series may also contribute to this); the indebtedness of the elements of the sample is characterized by a large standard deviation, and the average value fluctuates between 1.8 and 10.4.

The average indebtedness index of the examined Slovak SMEs shows a decreasing trend in the examined period, from 0.9 to 0.58 by the end of the period, which means that the value of the company's own capital exceeds the value of external capital in the company's capital structure (Figure 4D)

### *3.4. Efficiency*

Efficiency was expressed with the help of the Sales Revenue/Total Assets indicator. The cost-effectiveness of the use of resources (efficiency) in the case of the aggregated examined V4 SMEs ranged between 1.3 and 1.4 on average (in the years of 2012–2013, between 0.7 and 1.1), which means that the used resources (all assets) create the return that businesses expect from their use.

The efficiency of the examined Czech SMEs fell between 2014 and 2017, but even in the worst year, at least half of the enterprises produced an acceptable level of efficiency above 1.1. From 2017, an improving efficiency trend can be observed again.

The average of the annual efficiency of Polish SMEs between 2017 and 2021 ranges between 1.7 and 2.3; in addition, it can be said that half of the enterprises produce a value above 1.6–2.2 for each year examined, so half of the Polish health SMEs realized an average of at least 1.6–2.2 units of sales revenue by operating 1 unit of all assets. The Polish data do not show a uniform structure due to the small number of samples, but some of its elements show similarity to the Hungarian distribution (Figure 5B).

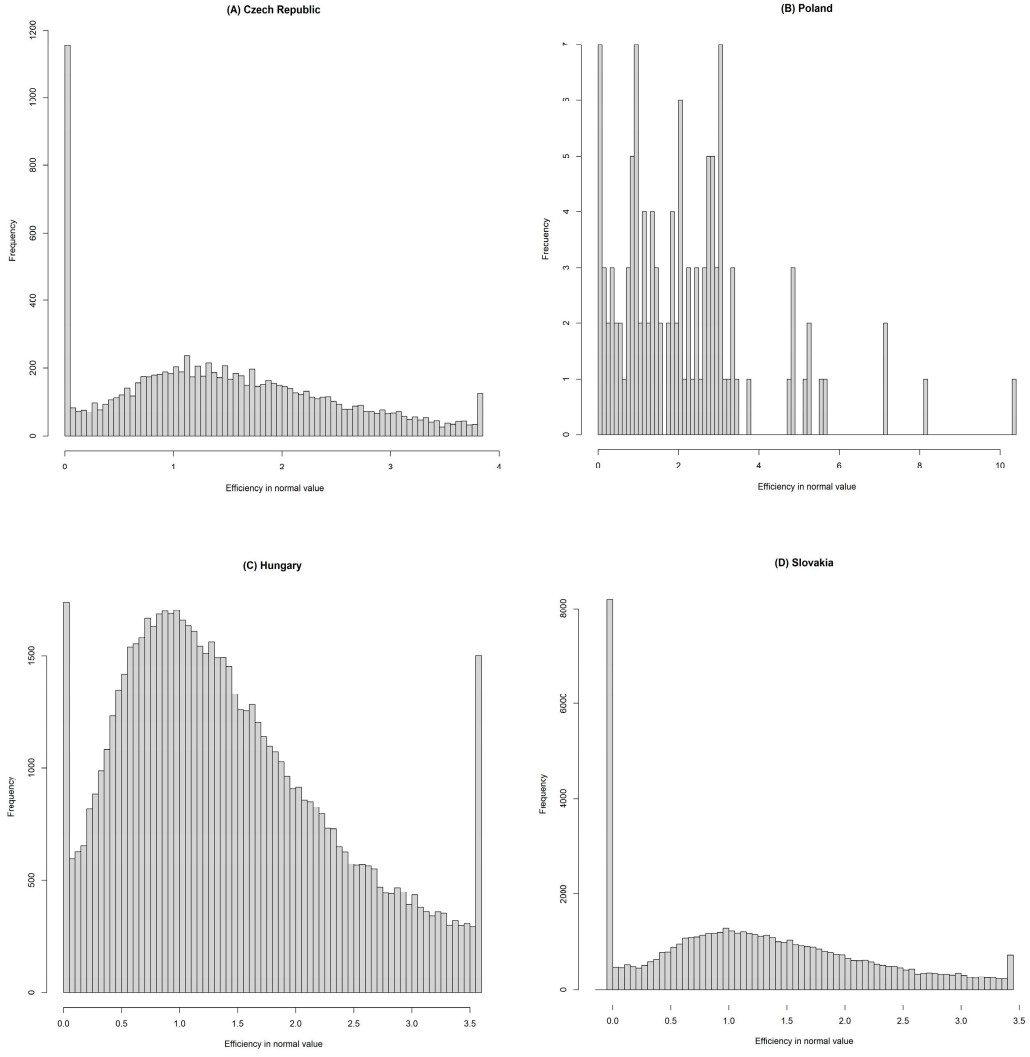

**Figure 5.** Efficiency histograms of the examined health industry SMEs in the V4 countries ((**A**) Czech Republic, (**B**) Poland, (**C**) Hungary, (**D**) Slovakia).

In the case of Hungary, we obtained balanced efficiency indicators with a low standard deviation, in which half of the analyzed SMEs show an efficiency above 1.2, and the average of the annual efficiencies ranges between 1.35 and 1.48.

In the case of Slovakia, we have data from 2013; according to our investigation, from 2014, the Slovakian SMEs operate with almost the same efficiency as the Hungarian ones.

It can be said about all four histograms that the location of the elements is not symmetrical; the patterns are positively skewed. The shape of the distribution of the Hungarian sample (Figure 5C) shows a negative peak, similar to the Czech (Figure 5A) and Slovak (Figure 5D) samples, which are flatter than the normal distribution, i.e., individuals are less clustered around the mean than we would expect in the case of a normal distribution; they have a larger standard deviation and take on higher extreme values.

*3.5. Pairwise t-Statistics for the Examined Financial Indicators*

Regarding the examined period, the pairwise *t*-statistics were prepared for all the indicators we examined, broken down by country pairs. We checked whether the difference between the pairs was significant, i.e., large enough to accept the null hypothesis. H0 means there is no significant difference between the averages of the examined financial indicators of the two countries. H1 means there is a significant difference between the averages of the examined financial indicators of the two countries. If the *p* value is above 0.05, then H0 can be accepted (marked with green color in Table 1 and Supplementary S1). If the *p* value is below 0.05, then H0 can be rejected and H1 accepted. The H0 hypothesis was verified most often in the case of the Slovak–Czech country pairs (indebtedness three times and ROA two times). In the year of 2021, the Polish–Czech and Polish–Hungarian country pairs proved to be significantly identical in the averages of all four examined financial indicators.

**Table 1.** Results of the pairwise *t*-statistics in the year 2021.

| | Pairwise *t*-Test *p* Value | | | | | | | | | | | | | | | |
|---|---|---|---|---|---|---|---|---|---|---|---|---|---|---|---|---|
| Year\Indicator | ROA | | | | Liquidity | | | | Efficiency | | | | Indebtedness | | | |
| | CZ | HU | PL | | CZ | HU | PL | | CZ | HU | PL | | CZ | HU | PL | |
| 2021 | HU | 0.0150 | | | HU | 0.0000 | | | HU | 0.0980 | | | HU | 0.0980 | | |
| | PL | 0.7950 | 0.9620 | | PL | 0.8928 | 0.1283 | | PL | 0.7500 | 0.9030 | | PL | 0.7500 | 0.9030 | |
| | SK | 0.0640 | 0.3070 | 0.9790 | SK | 0.0000 | 0.0003 | 0.2127 | SK | 0.0000 | 0.0000 | 0.8200 | SK | 0.0000 | 0.0000 | 0.8200 |

It can be concluded from the investigated, aggregated V4 health industry enterprises that their total assets provided a profit of around 10% in the examined period, that is, they operated with an approximate rate of return of 10%. It can be said that liquidity and profitability are closely related—they are in inverse proportion (excess liquidity can lead to profitability decrease and loss of profit); however, there are some industries where this is not the case (agricultural industry [77]). In our research, we proved that the health industry can also be classified as one of these sectors. If the requirements of liquidity (solvency) and profitability apply at the same time, then the sustainable growth rate of the enterprise is favorable, and its viability seems to be ensured in the longer term. But, for this, let us look at a survival estimator, the Kaplan–Meier survival estimator, and how this is fulfilled for the examined V4 health enterprises.

*3.6. Kaplan–Meier Estimator and Log-Rank Test*

The Kaplan–Meier estimator (KM estimator) enables the description of survival probabilities over time. The survival function shows how the percentage of survivors is expected to decrease over time from an initial event.

Based on the KM estimator, it can be concluded that the survival probability of the health industry enterprises in the examined countries is high. In the analysis, we modeled at least the 5-year survival for the V3 countries (without Poland) starting from 2010, for which the following modifications were made in the entire database: during the KM estimator setup, it was a basic condition to be an existing business in 2016 or before, and to have a business report for the given year. For companies that did not operate in the first place (equity and sales revenue = 0, and if the balance sheet total and equity

decrease from a positive value to 0 or below for the following year), we classified them as non-operating companies.

Based on the aggregated data of 6590 SMEs in the Hungarian KM estimator, only 27 businesses closed on average during their 1-year existence, which is 0.4% of all businesses. The death rate is 1.2% by the end of the 5th year, 2.5% by the end of the 10th year, and only 4.9% by the end of the 13th year. Figure 6C represents the KM estimator of the Hungarian sample, in which the black line shows the average, the proportion of survivors, and the gray bar shows the interval estimation, with a ±5% significance level along the average. We can see the angularity that results from the long time series. Evaluating what we have seen: the survival rate of the Hungarian health SMEs included in the study is extremely high, with a very low standard deviation. Therefore, it is worth starting and operating a healthcare SME in Hungary based on this analysis.

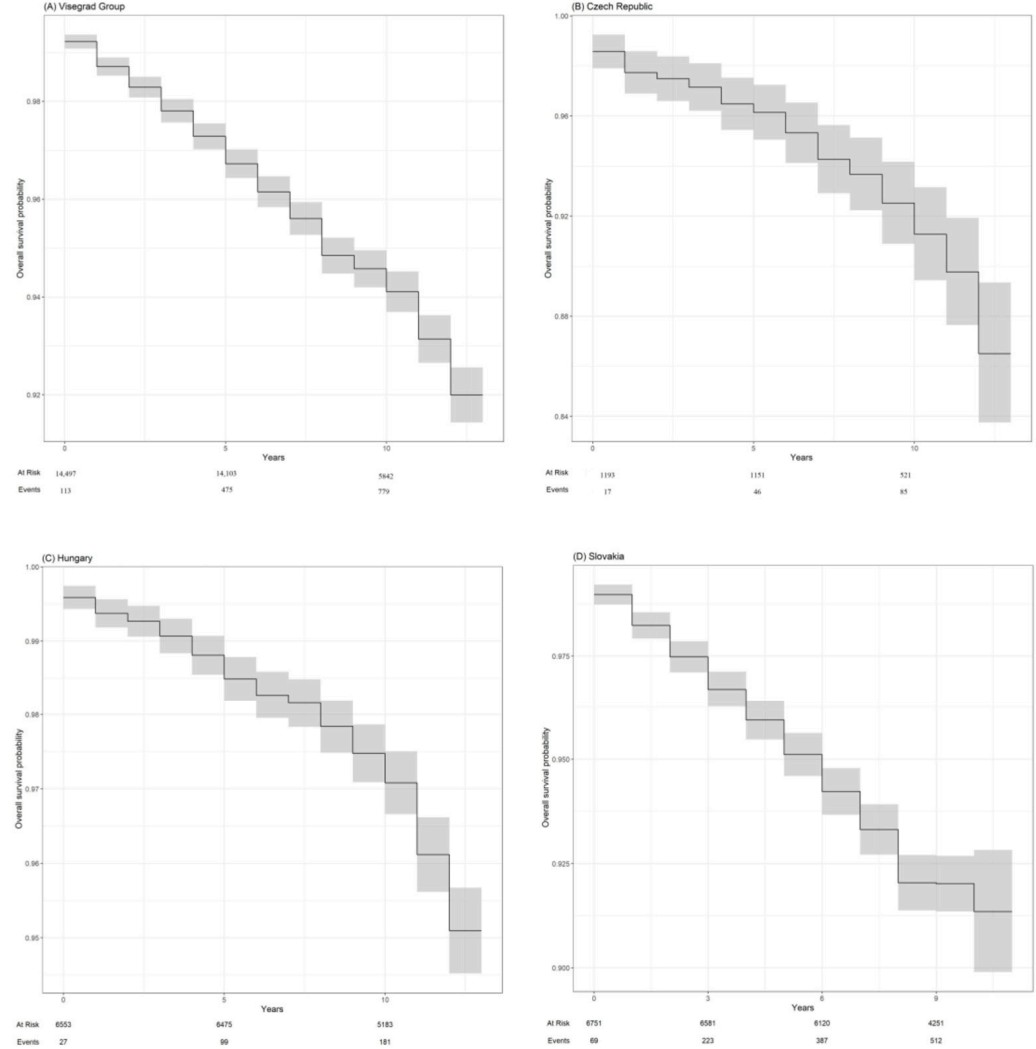

**Figure 6.** Survival estimator of the examined health industry SMEs in the V4 countries ((**A**) Visegrad Four, (**B**) Czech Republic, (**C**) Hungary, (**D**) Slovakia).

A total of 1210 SMEs were included in the Czech sample (Figure 6B). It can be seen that the deviation of survivors is increasing over the years. By the end of the first year, the average dropout rate is 1.4%, which is 3.5% by the end of the 5th year, 7.5% by the end of the 10th year, and 13.5% by the end of the 13th year (here, the deviation is already 1.4%).

The number of investigated Slovakian industrial enterprises (Figure 6D) is almost the same as the Hungarian one, 6820 in the starting year. On average, by the end of the 1st year, 1% of the individuals in the sample did not live/work, 4.1% by the end of the 5th

year, and 8% by the end of the 9–10th years. In the 11th year, the deviation of the interval estimate increases.

The KM estimator aggregating V3s is similar to the Hungarian KM estimator mostly (Figure 6A). Based on the overall averages, an average of 0.8% of all involved businesses did not survive the 1st year; 2.7% the 5th year; 4.6% the 10th year; and 8% the last examined year, the 13th year.

Interestingly, we can compare the KM estimator of Hungarian healthcare SMEs with the entire Hungarian company stock. Based on this, it can be said that the viability of SMEs in the examined sector (Figure 7B) is exceptionally good compared to the enterprises of the entire national economy (Figure 7A). Based on a rough estimate, only 85–90% of businesses are operational by the end of the first year, which is around 65% on average by the end of the 5th year, and close to 30% on average by the 10th year; after that, the proportion of viable enterprises continues to fall sharply.

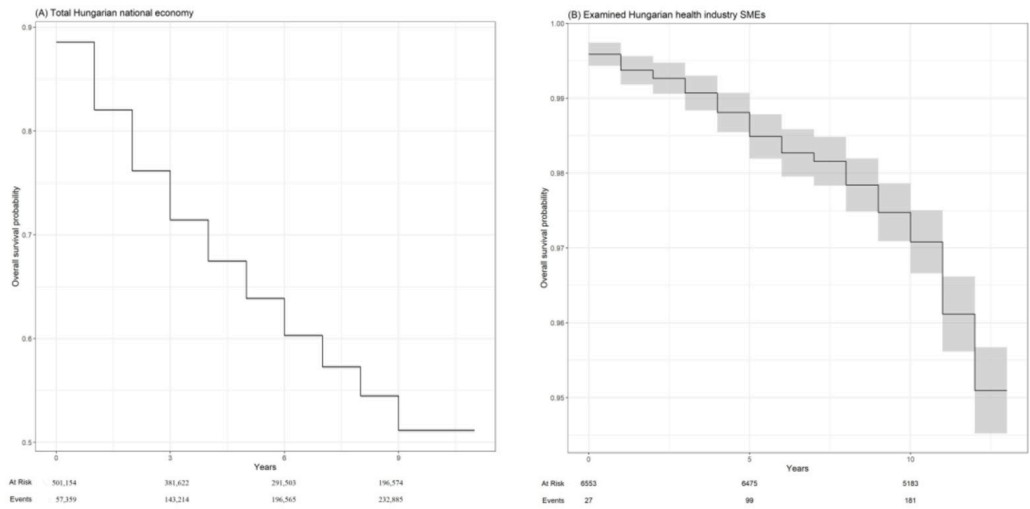

**Figure 7.** Kaplan–Meier survival estimator of Hungarian enterprises ((**A**) Total Hungarian national economy, (**B**) Examined Hungarian health industry SMEs).

The Kaplan–Meier survival curve shows the change in the survival rate among the examined companies over time, while the log-rank test statistically evaluates the difference among countries based on the survival curves, which helps to decide whether the differences among the examined countries are real or evolved by accident. The results of the log-rank test are presented in Table 2 and Supplementary S2. The Kaplan–Meier survival curves show similar results in the three examined countries; the survival rate of the health industry SMEs included in the study is extremely stable. Based on the log-rank test, we can make the following conclusions (Figure 8): there is a statistically significant difference in the survival of the investigated Hungarian–Slovak and Czech healthcare SMEs ($p \leq 5\%$ for all three countries). So, the three curves cannot be considered identical. The significance level of the survival rate of the analyzed Slovak and Czech healthcare SMEs is 5% (at the acceptance level). Although, based on the log-rank test, the survival rates in the three examined countries cannot be considered statistically the same, due to the economically very similar processes, the Hungarian–Czech–Slovak healthcare SMEs also have a survival rate of over 90%, based on our empirical results.

**Table 2.** Results of the log-rank test.

| Log-Rank Test | | | | | |
|---|---|---|---|---|---|
| | **Number** | **Observed** | **Expected** | $(O-E)^2/E$ | $(O-E)^2/V$ |
| Hungary | 6553 | 273 | 453 | 71.7 | 193 |
| Slovakia | 6751 | 513 | 333 | 97.6 | 193 |
| Chisq = 193 on 1 degrees of freedom, $p \leq 2 \times 10^{-16}$ ($p \leq 0.0000000000000002$) | | | | | |
| Hungary | 6553 | 273 | 331.8 | 10.4 | 90.3 |
| Czech Republic | 1193 | 103 | 44.2 | 78.4 | 90.3 |
| Chisq = 90.3 on 1 degrees of freedom, $p \leq 0.0000000000000002$ | | | | | |
| Slovakia | 6751 | 513 | 507 | 0.0742 | 0.498 |
| Czech Republic | 1193 | 103 | 109 | 0.3445 | 0.498 |
| Chisq = 0.5 on 1 degrees of freedom, $p = 0.5$ | | | | | |

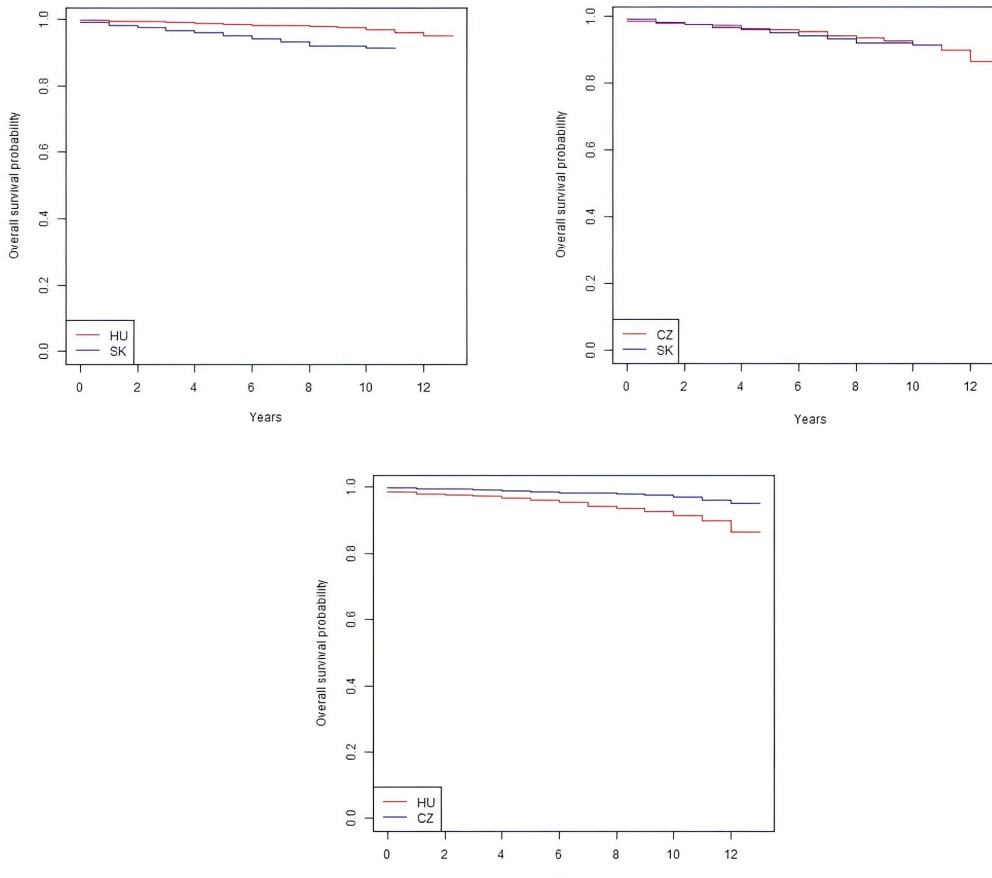

**Figure 8.** Validation of Kaplan–Meier survival estimator with log-rank test.

## 4. Discussion

We did not find any empirical research specifically dealing with the financial indicators of companies in the SME segment of the entire health industry in the V4 countries, with which we could compare our results. However, if we look at the sub-sectors or the entire corporate sector, there are already comparable studies. Regarding the profitability situation of the pharmaceutical companies in the Visegrad countries—return on equity (ROE)—it can be said that the financial risk of indebtedness and greater gearing within the industry mostly affects the already loss-making manufacturers, who are less financed by the bank, and more typically by private equity investors or internal financing sources [78]. Further

research is intended to reveal the financial effects of digitalization: the relationship between digitalization and financial performance was researched from 2017 to 2021 on a sample of healthcare companies from 12 European Union countries. Scafarto et al.'s (2023) results proved that the digitalization performance of companies is greatly influenced by the efficiency of capital use, while intellectual capital and human capital partly contributed to this [65].

In connection with the financial analysis of companies, researchers have highlighted several connections: Lang et al. (1996) found a negative relationship between leverage and growth [79]; we could not confirm this connection with our results. According to Mikula (1992), the interests of profitability and liquidity have equal value [80], while in the view of Borbély (1990), liquidity is more important than profitability in the short term, since liquidity does not preclude profitability, but insolvency can lead to bankruptcy, even in the case of profitable activities [81]. Other researchers have found evidence of a statistically insignificant relationship between leverage and the growth opportunities of SMEs, therefore, rejecting the assumptions of the Trade-Off theory. Their results showed a negative and statistically significant relationship between size and debt in their V4 SME sample. Regarding liquidity, their results are consistent with the Pecking Order theory, i.e., higher asset liquidity contributes to lower leverage, and company size has a negative and statistically significant effect on the capital structure [38].

Baumöhl et al. (2019), using the Kaplan–Meier survival estimator to examine the survival of businesses in the V4 countries between 2006 and 2015, concluded that the health industry (health and social work activities (Q)) is the second most viable industry after education. The entire period Kaplan–Meier survivor function was 0.915 [82]. In comparison with the results of the OECD, the 5-year survival rate of enterprises in the case of the V4 industry remains below 50% on average; in the case of Hungary, it is 42% [83]. Jung et al. (2018) proved that R&D investments can positively influence the probability of the survival of companies capable of producing innovative and intellectual property during a recession [84]. The results of Simón-Moya et al. (2016) show that new firms have a greater likelihood of surviving during crisis periods than during growth periods [85]. Our results obtained during the Kaplan–Meier survival analysis are consistent with the results of Baumöhl et al. (2019) and OECD (2016). Based on our empirical analysis, we proved that the SME segment of the health industry in the examined countries shows fruitful financial results, in contrast to the eroding healthcare system [29,86,87] and the poor health status of the population. This study also contributes to research connected to the healthcare entrepreneurship and financial performance. In particular, it enriches the empirical literature of health economics, establishing the importance of SMEs in the health industry in this extremely knowledge-intensive and technology-driven sector. Our research makes a significant contribution to the international literature, as it fills a significant gap in the healthcare SME segment of the V4 countries. Although there are already existing comparable studies on certain sub-sectors of the healthcare sector or the entire corporate sector, the analysis of the financial indicators of the entire healthcare SME segment is still missing in the context of the V4 countries. By applying financial analyses and statistical models, the research covers the entire SME sector, and not only the main subsectors, thus providing a complex picture of the entire healthcare SME segment in the V4 countries. Unlike studies found in the international literature, our research results approach the urgent transformation of the health policy of the V4 countries from a new perspective, as we found that the profitable, efficient, and stable performance of the examined health industry SMEs (based on the main financial indicators and survival analysis) is combined with the health status of the population of the V4 countries, which is below the OECD average. Among the possible reasons may be that, due to the shortcomings of the state healthcare system, a profitable and stable private health industry base serving the needs of the population and patients at a higher level has been built up; the health awareness of the population of the V4 countries is low, and the level of prevention is low compared to patient care and treatment. Mapping the causes of the connection is a complex task, but it can create an opportunity to

create a health industry ecosystem, in which the health standards and well-being of the population will increase. The results can, thereby, support the understanding of internal dynamics, correlations, and trends within the health industry sector. This added value is fundamental to the international literature, as it provides a broader context and new directions for further research.

## 5. Conclusions

Based on Statista's industry reports, it can be concluded that the overall health industry performance of the V4 countries increased (+18%) compared to the average of the 3 years before the COVID-19 pandemic in the average of the first 3 years of the COVID-19 pandemic; however, for the performance of individual sub-sectors and the influencing factors, examination of root causes is essential to establish correct conclusions and trends. A key aspect of our study was the analysis of the life expectancy at the time of the founding of healthcare SMEs operating in the V4 countries. For the sake of long-term stability and sustainability, it is extremely important to understand the survival chances of businesses in the post-launch period. During this analysis, we used the Kaplan–Meier survival estimator, which enabled us to compare the lifespan of businesses and identify differences among different countries. According to our results, the life expectancy of healthcare SMEs operating in Hungary at the time of establishment is significant, i.e., they show a high survival rate, which indicates that these enterprises are sustainable and stable in the long term. However, interestingly, the high survival rate is not unique among the countries studied. In our analysis, no difference can be detected between Hungarian healthcare enterprises and the healthcare SMEs of the other examined countries, which indicates that the high survival rate is a general characteristic of the healthcare SMEs of the V4 countries. A multifaceted examination of the healthcare SMEs of the Visegrad countries—analysis of the solvency (liquidity), profitability (ROA), indebtedness (capital structure), and efficiency ratio, supplemented with the Kaplan–Meier survival estimator—provides sufficient grounds to state that the examined SMEs do not need financial support; they stand on stable feet, which provides them an excellent opportunity to either expand abroad or introduce innovations. Furthermore, we verified that the management of the examined health industry SMEs in the V4 countries differs from the general management of enterprises on several points (liquidity–ROA relationship, survival analysis). Based on the examined financial indicators, the SMEs of the countries included in the study are liquid, they have stable solvency, and they operate efficiently; so, it can be said that it is worth investing in this industry, since there is a high chance that a new enterprise (which is not necessarily a startup) will be predestined for success.

Compared to the OECD member states, the health system performance of the V4 countries falls short of the desirable level. The strong exposure of the population to health risks, as well as the shortcomings of the healthcare system, contribute to the current underdeveloped standard of health compared to other countries. Among the life qualities of the population of the V4 countries, increasingly large differences and deviations can be found in the field of health and the quality of life indicators. The latter is confirmed by the OECD's annual health country profiles [88–92]. The demand for private healthcare providers and the promotion of better health and disease prevention are increasing. Drawing a parallel between the living standards indicators of the V4 countries and the health industry SMEs of the examined countries, we can establish that it is highly probable that the disease treatment—and not the health understanding and development and prevention industries—perform outstandingly.

The sustainable development supposes a high quality of life and its preservation [93], coupled with long-term value creation from economic development [94]. The healthcare systems of the V4 countries have been underfunded for a long time, and the reforms focusing exclusively on economic goals and measures aimed at long-term sustainability are lacking [29].

All in all, it can be said that the data from Statista (which examines the entire corporate sector, not narrowed down to SMEs) and our financial analysis from the Crefoport database show that the SME sector of the health industry is flourishing in V4 countries. The reason for this successful operation in the member states of the Visegrad countries is that the population has limited access to public healthcare (for example, mainly due to long waiting lists and the lack of specialized staff) [95–98], so those who can afford it try to use private health services.

## 6. Future Research and Limitations

Compared to the V3 (Czech, Slovak, Hungarian) countries, the number of Polish enterprises included in the study is lower and covers a shorter time frame (due to the characteristics of the Crefoport database); for this reason, our results cannot be considered representative for Poland. Later on, a representative study can provide further clarifications/confirmations.

Due to the aggregated data covering the sector as a whole, the changes and trends that occurred in certain industries of the health industry in the given years cannot be proved in our research; a more detailed analysis of these can form the basis for further research. Further research questions can be raised as to what proportion health promotion, disease prevention, and the treatment of already established diseases contributed to the impressive performance of the health industry SMEs of the examined countries. In the present study, we did not examine the proportion of public and private sector representation, which would provide additional relevant explanations.

An analysis based on this research can be continued in the future to verify the relationship between the survival of healthcare SMEs and the shortage of doctors. Probably, the general shortage of doctors results in the stable operation of healthcare SMEs. In other words, the demand is much higher than the supply; in this case, the number of patients is much higher than the number of doctors needed.

## 7. Research Implications

Our research is significant from several points of view in the examination of the health industry SMEs of the CEE region, especially with regard to the V4 countries. First of all, our results point to the important role that these SMEs play not only in the health industry, but also in the economy as a whole. The particularly high life expectancy shown by health industry enterprises operating in the countries included in the study is an important indicator that these enterprises operate stably and sustainably, are able to respond to changes in the industry, and adapt to changing conditions. Second, our study can contribute to policymaking by identifying areas where additional support and regulation may be needed for the further development of healthcare SMEs. The effects of Industry 4.0 and the COVID-19 pandemic are particularly important, as they represent a significant challenge for these businesses, but at the same time, they create new opportunities for digitization and innovation. Last, but not least, our study also highlights that healthcare SMEs play a significant role in promoting social welfare, as their services make healthcare accessible and contribute to the efficient use of healthcare system resources. This means that the further development and success of businesses in the healthcare industry directly contribute to the quality and accessibility of healthcare in society, which is key to the well-being of society as a whole.

The results of our study contribute to a deeper understanding of the role and importance of healthcare SMEs and can guide policymakers, company managers, and the scientific community on what steps they can take in the future to promote the further development of healthcare SMEs.

**Supplementary Materials:** The following supporting information can be downloaded at: https://www.mdpi.com/article/10.3390/su151612360/s1, Supplementary S1. Pairwise *t*-test *p* value. Supplementary S2. Basic data.

**Author Contributions:** Conceptualization, J.V.-D. and L.P.; methodology, B.V. and J.V.-D.; software, B.V.; validation, A.D., L.P. and J.V.-D.; formal analysis, G.T.; investigation, J.V.-D.; resources, J.V.-D.; data curation, J.V.-D., L.P. and B.V.; writing—original draft preparation, J.V.-D., L.P. and A.D.; writing—review and editing, G.T.; visualization, B.V.; supervision, L.P. and A.D.; project administration, G.T.; funding acquisition, A.D. All authors have read and agreed to the published version of the manuscript.

**Funding:** This research received no external funding.

**Institutional Review Board Statement:** Not applicable.

**Informed Consent Statement:** Not applicable.

**Data Availability Statement:** The data presented in this study are available on request from the corresponding author.

**Conflicts of Interest:** The authors declare no conflict of interest.

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
