# Peer review of "Financial Analysis and Survival Research of the Visegrad Countries’ Health Industries"

_sustainability, doi:10.3390/su151612360_

Round 1
Reviewer 1 Report
In the article, the authors conducted a comprehensive financial analysis of the health industry of the Visegrad countries. The analysis was performed on the basis of the values of classical indicators characterising the financial health of companies. In addition, the study used survival analysis methods – the Kaplan-Meier estimator.
Consideration of the following comments by the authors may help to improve the quality of the article:
1. The introduction should describe the purpose of the article. At the moment, the purpose is only specified in the abstract. In addition, it would be useful in the introduction to describe the layout of the article.
2. There is a general description of the research hypothesis (called the main hypothesis by the authors) in the Abstract. There is a lack of precise formulation of the hypothesis (or hypotheses) in the Introduction or in the following sections.
3. The authors use the term Kaplan-Meier model. Classical textbooks on survival analysis use the term Kaplan-Meier estimator.
4. Please explain exactly why the Kaplan-Meier estimator was not used to analyse the survival of companies in Poland.
5. Please provide the formula for the Kaplan-Meier estimator and describe exactly what the random variable T describes and what is the initiating event of the observation and what is the terminating event of the observation (probably the bankruptcy of the company). Describe exactly what was taken as a censored observation and how many censored and full observations there were. Readers familiar with survival analysis may guess at some assumptions. Conversely, readers unfamiliar with these methods will not know what was actually investigated in this section.
6. The authors compared survival curves for companies in the individual Visegrad countries. However, they did not use any tests for this. Duration curves can be compared using: log-rank test, Gehan test or others. I recommend the article: Karadeniz, Pinar Gunel, and Ilker Ercan. "Examining tests for comparing survival curves with right censored data." Stat Transit 18.2 (2017): 311-28. https://sit.stat.gov.pl/Article/398.
7. The conclusions should address the research hypotheses.
English language needs minor proofreading.
Author Response
Dear Reviewer!
Thank you for your comments and suggestions, on the basis of which we were able to improve our study. We answer the points in the proofreader's review in the attached document!

Reviewer 2 Report
Authors should support the work with statistical evidences.

Author Response

(The authors gave the same response as above.)

Reviewer 3 Report
· The title of the paper should be modified, especially the second part. Also, adding a comma is not relevant.
· Abstract is too long. It should be shortened and made clear to the readers. Please stick to the main purpose of the study, method used, results and conclusion.
· The first sentence in introduction requires a reference.
· The supporting theory for this study should be added in the literature review section.
· Some paragraphs are too long, which makes it difficult for the readers to follow through (see page 3).
· Research implications (theoretical and practical) cannot be found anywhere in the paper. They should be added.
· The title of the paper should be modified, especially the second part. Also, adding a comma is not relevant.
· Abstract is too long. It should be shortened and made clear to the readers. Please stick to the main purpose of the study, method used, results and conclusion.
· The first sentence in introduction requires a reference.
· The supporting theory for this study should be added in the literature review section.
· Some paragraphs are too long, which makes it difficult for the readers to follow through (see page 3).
· Research implications (theoretical and practical) cannot be found anywhere in the paper. They should be added.
Author Response

(The authors gave the same response as above.)

Round 2
Reviewer 3 Report
Dear Editor,
Thank you for the opportunity to review the paper. Although the author responded to my comments, but some of them are not well addressed. First, comma should not be included in the title and some modifications are required. Second, the problem statement requires strengthening. Third, theoretical implications are not presented in the paper. the author should discuss how this study will advance the theory and literature. Finally, there are many grammatical errors and issues in the writing. In some parts, the author mentioned "this document", in others it mentioned "this paper". Also, in line 116 (page 3), the author stated "In this chapter we collected studies about the health situation of the Visegrad countries for further examination". This is not a chapter to mention it.
Proofreading is required.
Dear Editor,
Thank you for the opportunity to review the paper. Although the author responded to my comments, but some of them are not well addressed. First, comma should not be included in the title and some modifications are required. Second, the problem statement requires strengthening. Third, theoretical implications are not presented in the paper. the author should discuss how this study will advance the theory and literature. Finally, there are many grammatical errors and issues in the writing. In some parts, the author mentioned "this document", in others it mentioned "this paper". Also, in line 116 (page 3), the author stated "In this chapter we collected studies about the health situation of the Visegrad countries for further examination". This is not a chapter to mention it.
Proofreading is required.
Author Response

(The authors gave the same response as above.)
